# A System Dynamics Model for Assessing the Efficacy of Lethal Control for Sustainable Management of *Ochotona curzoniae* on Tibetan Plateau

**Jiapeng Qu** [1,2,*], **Zelin Liu** [3,4], **Zhenggang Guo** [5], **Yikang Li** [1,2] and **Huakun Zhou** [1,2]

[1] Key Laboratory of Adaptation and Evolution of Plateau Biota, Northwest Institute of Plateau Biology, Chinese Academy of Sciences, Xining 810008, China; ykli@nwipb.cas.cn (Y.L.); hkzhou@nwipb.cas.cn (H.Z.)
[2] Qinghai Provincial Key Laboratory of Restoration Ecology in Cold Region, Xining 810008, China
[3] College of Resources and Environmental Science, Hunan Normal University, Changsha 410081, China; zelin.liu@hunnu.edu.cn
[4] Department of Biological Sciences, University of Quebec at Montreal, Montreal, QC H3C 3P8, Canada
[5] State Key Laboratory of Grassland Agro-Ecosystems SKLGAE, Lanzhou University, Lanzhou 730000, China; guozhg@lzu.edu.cn
[*] Correspondence: jpqu@nwipb.cas.cn

**Abstract:** When population abundances exceed the economical or ecological threshold, animals are always regarded as pests, so effective and sustainable management strategies are required. As a native species widely distributed on Tibetan Plateau, plateau pika is regarded as a pest when its abundance is too high. For decades, plateau pika were controlled using toxic baits by both the local government and shepherds. However, how pika population fluctuates after lethal control is far from certain. Based on our previous studies, demographic parameters of plateau pika were acquired. A system dynamics model is developed to parameterize a population model for this species. The model incorporates two age categories (juvenile and adult) of both sexes, and uses density-dependent factors, including reproduction, mortality, and migration. The model is used originally to analyze the effect of pika management on the population dynamics and resulting abundance, in order to plan optimal pest controlling strategies. The results show that lethal control is efficient when continuously conducted once each year, or twice with 2-year intervals. For sustainable controlling pest abundance, comprehensive strategies should be considered. An appropriate population model could be used to explore the optimal strategies and provide important reference into decision making about pest management.

**Keywords:** alpine meadow; rodent; pest management; plateau pika; population dynamics



## 1. Introduction

When animal abundance exceeds the economical or ecological threshold, or the animal invades into a new habitat, causing adverse effects to the ecosystem, it is regarded as a pest [1]. Pest control is one of the global hot topics for health, economy, and ecology [2]. Rodent pests transmit diseases to humans and domestic animals, graze crops and grasses, and affect ecosystem diversity [3]. The most popular approach for pest control is lethal control using pesticides, which leads to the acute toxicity and death of pests and rapid abundance decline [4]. However, the pest population always recovers or even exceeds the natural abundance, thus, sustainably controlling pest abundance over a long-term period is crucial for its management. Indeed, both reproduction and survival are density-dependent in the population regulation process [5]. After poisoning, residual or surviving individuals have more food and space resources, may reproduce with more litters and/or litter size, enhancing survival rate, which is termed "over-compensatory population growth" [6].

The population dynamics model is an important tool for predicting population fluctuations, and widely used to plan effective conservation strategies for endangered species [7].

Here, we attempted to build a system dynamics model for a rodent pest inhabiting the Tibetan Plateau. Through the analyses of the model, we can determine the factors limiting the population dynamics with abundance, assess the efficiency of management measures, and provide optimal management strategies for the decision maker [8].

The Tibetan Plateau, knows as the third pole of the earth, holds the largest alpine meadow ecosystem and provides a unique environment for a wide range of plateau biota. In the last decades, around 40% of the grassland has been degraded [9]. Grassland degradation influences the life of local pastoralists who rely on the grassland, and affects the ecological environments and wildlife. Rodent activity is regarded as one of primary factors leading to the grassland degradation [10,11]. There are around 800 million small rodents belonging to about 45 species on the Tibetan Plateau, which annually consume more than 1.5 million of fresh grass [12].

Plateau pika (*Ochotona curzoniae* Hodgson, Lagomorpha Ochotonidae) is widely distributed on the alpine meadow, representing over 75% of rodents of the Tibetan Plateau [13,14]. They are social mammals, and the social unit is a family with 2–3 adults and their offspring [15]. Plateau pika produce 1–5 litters per year with the mean litter size of 3–4, and its abundance can reach more than 380 individuals per ha [16]. It is regarded as a keystone species on the Tibetan Plateau since it provides nests for small birds (e.g., snowfinch), prey for predators such as Tibetan fox and upland buzzard [17,18], and its activity accelerates the materials circulation and improves the soil nutrient concentrations [19–21]. However, when its population is over abundant, plateau pika is considered as a pest because it competes for scarce food resources with livestock, and its grazing and digging decrease the vegetation diversity, aggravating the degradation of alpine meadow [11,19,22,23]. The control campaigns against plateau pika have been conducted using poisons such as Zinc phosphide from 1960s. Since the 1980s, anticoagulants, such as diphacinone-Na, or botulin toxin, were used to control the rodent populations on the Tibetan Plateau [3]. Under the support of the first and second stage project of "Ecological conservation and restoration project in Sanjiangyuan region", by 2014, around 400 million Yuan (61.5 million USD) were used to poison rodents, with plateau pika as the major target [24]. The poisoning project is always conducted between October to April, once or twice per year, which primarily depends on the budget of local governments. Though acute poisons can lead to an immediate decrease of pika abundance, its population recovers quickly in a few months [3,25]. It is urgent to explore the mechanism of the recovering of pest population after lethal control, and formulate suitable measures for sustainably reducing pest abundance [26].

In this study, we use capture–recapture and dissecting experiments to obtain the basic demographic parameters in both natural and lethal control populations. We are more interested in the population changes in the warm season, since the plateau pika reproduces and migrates, and the population fluctuates dramatically during this period [27]. System dynamics models are constructed to simulate population fluctuations under natural or lethal control scenarios. The model incorporates reproduction and mortality as variables, which are both density-dependent. The model predicts the population fluctuations under different measures, and contributes to planning optimal strategies to manage rodent populations. The objective of this study is to assessing the efficacy of different poisoning scenarios on controlling plateau pika abundances via the constructed system dynamics model.

## 2. Materials and Methods

### 2.1. Study Area

The study area is located at the southeast of Maqin County, Qinghai, China (34°24′ N, 100°21′ E, 3800–4000 m). This region experiences an arid climate. There are only cold (September to April) and warm (May to August) seasons. About 80% of the precipitation occurs from June to September, and the annual mean precipitation ranges from 380 to 630 mm. Snow usually begins in September and ends in June, and heavy snow packs rarely occur over winter. The daily changes in the temperature are relatively large (up to 25 °C),

whereas the average annual temperature is approximately 0 °C. Frost can occur throughout the year, and soil can freeze to a depth of >2 m during winter.

The vegetation at the study site is typical *Kobresia humilis* grassland, which is grazed by yaks (*Bos grunniens*) and sheep (*Ovis aries*) throughout the year. The common predators of plateau pikas include red foxes (*Vulpes vulpes*), upland buzzard (*Buteo hemilasius*), and large-billed crow (*Corvus macrorhynchos*).

### 2.2. Natural Population Demography

The natural population study site of 4.0 ha was located at the flat base of the mountains. We did the mark-recapture experiment from 2005 to 2009 to investigate the population demographic traits. The population dynamics and survival rate were obtained [16,27].

A two-years dissection experiment was conducted to measure the reproductive parameters of plateau pika at about 3 km away from the natural mark–recapture study site. Twenty adult female pikas were captured and dissected twice per month from May to August in 2007 and 2008, and litters and litter size were acquired [28].

### 2.3. Lethal Control Population Demography

The lethal control of plateau pika was conducted by the Grasslands Station of the Guoluo State in January 2007. Botulin toxin D was used to poison pikas. The total poisoning area was around 20.0 ha. The distance between the natural and poisoning study sites is about 12 km. We chose two sampling sites in this area. In one 5 ha site, ten lines with 100 m length and 10 m width for each were randomly chosen to investigate population density using line-transect method [29]. About 5 km away from this site, another 5 ha site was used to assess the reproductive parameters (i.e., litters and litter size) after lethal control. Twenty adult females were captured and dissected twice per month from May to August in 2007 and 2008 [30].

### 2.4. Model Structure and Parameterization

The pika model is a demographic simulation with variable pest management strategies, built in a system dynamics framework (STELLA, Version 10.0.3) (Figure 1, Table S1). It makes extensive use of re-sampling from recorded databases and of stochastic variability within configurable bounds.

The model distinguishes two age classes: Juvenile and adult, juveniles which are born in May (first litter) or June (second litter) become adults in August. The model uses the month as a time step. The main dynamics of the model are monthly reproduction, mortality, and migration. According to the study of Qu (2011), the environmental carrying capacity is assumed to be stable (250 ind/ha). Monthly mortality is stage-specific and density-dependent. The migration is regulated by population density and environmental carrying capacity. A model year runs from January (index 1 at initiation, 13 in the following years) to December (12).

Mortality rate for each age class and month are based on initial estimates by Qu et al. (2017). Reproductive parameters of natural and lethal control population were obtained by Qu et al. (2011, 2012). All parameters used in the model are described in Table S2.

At model initialization, each individual in the starting population is assigned to the adult age class, as plateau pikas begin to reproduce in May. The number of adult pikas is set at an initial chosen amount at the beginning of the first year of simulation (January or time 1). The model then calculates how many pikas are in each age group by assuming the population is initially in equilibrium (see the details in Table 1).

In the Guoluo region, plateau pikas reproduce in May and June, termed first and second litters, respectively. The initial number of pika in January (month = 1) is 33, which is coincident with the normal density in the natural population. The number of juveniles is the product of multiplying the adult females in breeding season by the mean litter size.

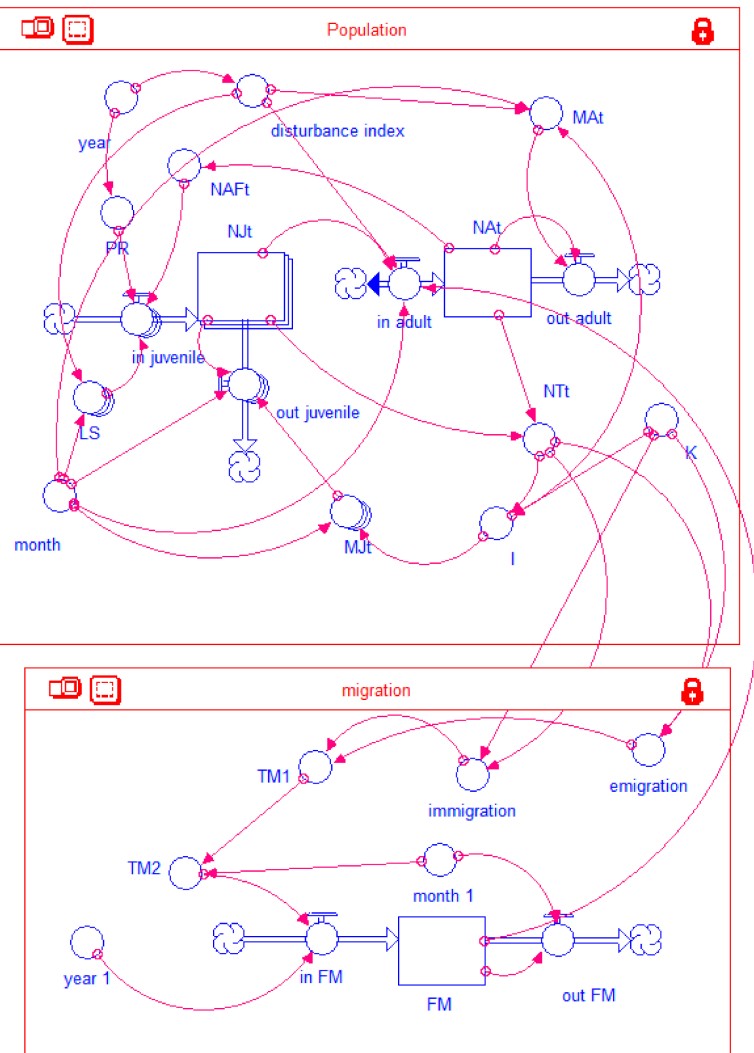

**Figure 1.** Conceptual diagram of a plateau pika population model. Note: Boxes indicate state variables (stocks), circles indicate driving variables, constants, or auxiliary variables, and arrows going from a state variable to another state variable with a circle touching the arrow are material transfers. NJ: Juvenile population, NA: Adult population, NT: Total population, MA: Adult mortality, MJ: Juvenile mortality, NAF: Adult female population, I: Index of death, K: Carrying capacity, LS: Litter size, FM: Number of final migration, PR: Pregnant ratio, disturbance index: Poison scenario, t: Different month.

In general, the poisoning program is carried out once or twice a year, consecutively or occasionally. We assume several scenarios: (1) Case 0, natural population without treatment; (2) case 1, pikas are poisoned in April with a 1-year interval; (3) case 2, pikas are poisoned in April with a 2-year interval; (4) case 3, pikas are poisoned in April with a 3-year interval; (5) case 4, pikas were poisoned in October with a 2-year interval, (6) case 5, pikas were poisoned in both April and October with a 2-year interval. We use the structured system dynamics model to simulate the population dynamics in 10 years. One-way ANOVA was used to compare the variations between simulated and observed population dynamics from 2005–2009, to evaluate the accuracy of the model. General linear mixed model (GLM), with treatment, year, and month in the year as mixed factors, is used to compare the differences of population density under different treatments.

**Table 1.** The description of algorithms in the plateau pika population model.

| |
|---|
| In juvenile = NAFt * PR * LS[L1] + NAFt + 1 * PR * LS[L2]. |
| Jt = in juvenilet − our juvenilet |
| NAFt = Nat * 0.51 |
| Out juvenile (L1) = if month = 8 then NJt[NL1] else NJt[NL1] * MJt[NL1] |
| Out juvenile (L2) = if month = 8 then NJt[NL2] else NJt[NL2] * MJt[NL2] |
| In adult = if month = 8 then (NJt[NL1] + NJt[NL2]) * 0.9 ELSE |
| IF month = 5 OR month = 6 THEN 0 ELSE |
| IF month = 7 THEN FM |
| ELSE 0 |
| Out adult = Mat * NAt |
| MA (Aug. to Apr.) = 0.04 * I |
| I = if NTt < K then 1 else 3 |
| NTt = Nat + NJt [NL1] + NJt [NL2] |
| NAt = 33 |
| NJt = 0 |
| Immigration = if NTt < 88 and NTt ≥ 70 then NTt * 0.3 ELSE |
| if NTt < 70 THEN NTt * 1 ELSE |
| if NTt > K then NTt * 0.01 ELSE NTt * 0.05 |
| Emigration = if NTt < 88 then 0 ELSE |
| if NTt > K THEN NTt * 0.1 ELSE NTt * 0.05 |
| TM1 = immigration-emigration |
| TM2# = if month_1 = 1 OR month_1 = 2 OR month_1 = 3 OR month_1 = 4 |
| THEN TM1 ELSE 0 |
| Month 1 = time MOD 12 |
| Out FM = IF month_1 = 8 |
| THEN FM ELSE 0 |

## 3. Results

### 3.1. Plateau Pika Population Trend under Natural Conditions

The simulated population density showed seasonal population dynamic trends consistent with the observed natural population in the warm season from 2005 to 2009 ($F_{1,17}$ = 46.066, $p$ < 0.001). Following the reproduction of pikas, the density increased in May and June, reached the peak in July, then declined quickly.

### 3.2. Model of Plateau Pika Population

Briefly, the stable population density fluctuation appeared since the third year, then showed similar monthly dynamic patterns in the following years (Figure 2). In other words, the model could run stable from the third year. Thus, we primarily compared the population dynamics under different scenarios since the third year.

Case 0:

Figure 2 showed that in the natural scenario, the simulated density ranged from 118.2 to 276.6 ind/ha in the warm season, while it ranged from 80.7 to 112.2 ind/ha in the cold season (Figure 2, case 0).

Case 1–3:

When pikas were poisoned in April with a 1-year interval (Case 1), the densities were significantly lower than those of natural population. However, when pikas were poisoned in April with a 2- (Case 2) or 3-year interval (Case 3), no significant differences of density were detected compared with Case 0 (Figure 2, Table 2).

Case 4–5:

When pikas were poisoned in October with a 2-year interval (Case 4), the densities were not significantly different than those of Case 0. However, when pikas were poisoned in both April and October with a 2-year interval (Case 5), the densities were significantly lower than those of Case 0 (Figure 2, Table 2).

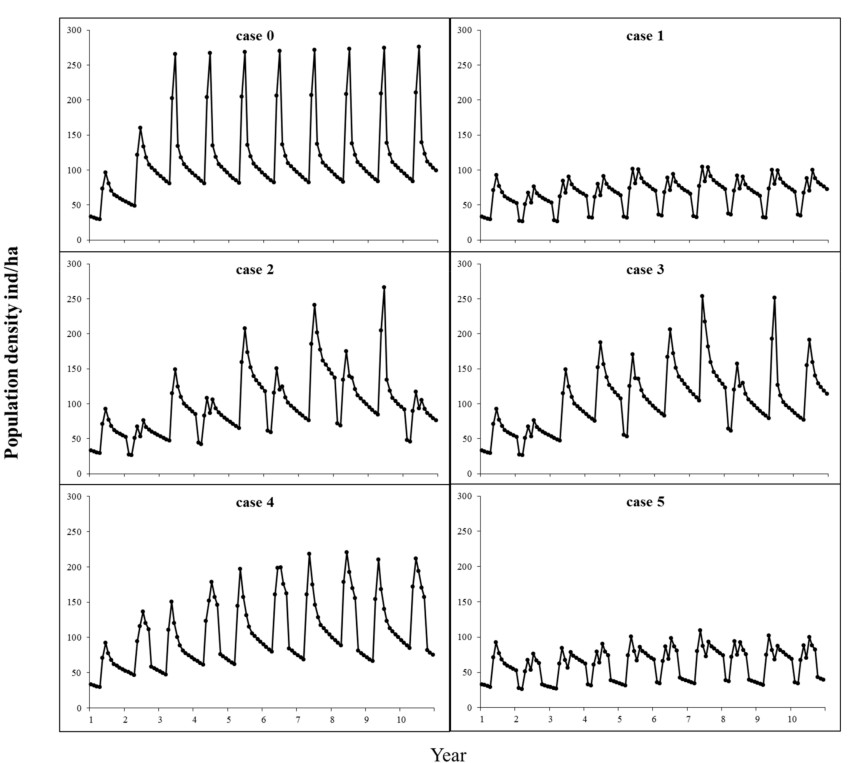

**Figure 2.** Result of model simulation of population dynamics in plateau pika.

**Table 2.** Summary of general linear mixed model (GLM) analyses of density variation among different poisoning scenarios. Significant *p*-values are in bold font.

| Source of Variation | Estimate | SE | *t* Value | *p* Value |
|---|---|---|---|---|
| Intercept | 120.582 | 12.024 | 10.028 | **<0.001** |
| Case 1 | −56.527 | 17.005 | −3.324 | **0.001** |
| Case 2 | −26.875 | 17.005 | −1.580 | 0.115 |
| Case 3 | −19.733 | 17.005 | −1.160 | 0.246 |
| Case 4 | −27.388 | 17.005 | −1.599 | 0.108 |
| Case 5 | −62.887 | 17.005 | −3.698 | **<0.001** |
| Year | −0.687 | 1.801 | −0.381 | 0.703 |
| Year: Month | 0.210 | 0.069 | 3.062 | **0.002** |
| Case 1: Year | 0.310 | 2.467 | 0.126 | 0.900 |
| Case 2: Year | 1.626 | 2.467 | 0.659 | 0.510 |
| Case 3: Year | 2.010 | 2.467 | 0.815 | 0.416 |
| Case 4: Year | 4.175 | 2.467 | 1.692 | 0.091 |
| Case 5: Year | 0.307 | 2.467 | 0.124 | 0.901 |

Note: Case 1 to Case 5 mean different plateau pika poisoning scenarios. Case 1: Poisoning pikas in April with a 1-year interval; Case 2: Poisoning pikas in April with a 2-year interval; Case 3: Poisoning pikas in April with a 3-year interval; Case 4: Poisoning pikas in October with a 2-year interval; Case 5: Poisoning pikas in April and October with a 2-year interval.

## 4. Discussion

In this study, a system dynamics model was built and well simulated the population dynamics of plateau pikas. Considering its wide distributions and high abundances, the need for effective management measures to reduce plateau pikas is urgent [11]. The pika density could recover when conducting the poisoning program with a 2- or 3-year interval, as has been observed in previous studies [25,31]. Only continually poisoning pikas could effectively control pika abundance over long term period.

### 4.1. Model Evaluation

The system dynamics model can integrate the surrounding environment and potential human activity, and thus has been widely used in the management and conservation of fishes, birds, and mammals [32–34]. Our model appears to well describe the population dynamics of plateau pikas. Simulated populations showed similar annual and seasonal fluctuations of plateau pikas in this region [27].

The reproduction of plateau pikas primarily occurs in May and June, with stable litters and litter size [28]. Here, we did not consider the density-dependent reproduction. It should be noted that the reproduction of plateau pikas shows geographic variations, as females produce more litters with higher litter sizes at low altitudinal regions [35,36]. When using this model to simulate population dynamics in other regions, the variable reproductive parameters should be considered.

The density-dependent survival is integrated into the pika model, limiting the population fluctuations. Density-dependent survival can affect animal population dynamics, especially when their abundances are high in winter [37,38]. However, the survival rate of plateau pikas is high in the cold season, but low in the warm season [16]. Thus, a dramatic population fluctuation is observed in the warm season. We considered the effects of predation risk on the survival in the model. As the survival rates used in the model were obtained from the natural population, the predation risk reflected the real natural condition [16]. We recommend to incorporate the predation effects to the model to better simulate the population dynamics in the future, especially for regions with variable predation risks.

Both immigration and emigration were considered in the model. For the open natural population, migration is a crucial factor influencing population dynamics [39]. The migration of plateau pikas primarily occurs before the starting of the breeding season (i.e., from January to April) [35]. Here, we assumed that, when the population density was less than 88 ind/ha, there was primarily the immigration; when the density was between 88 and 250 ind/ha, the immigration and emigration were equal, i.e., the net migration was zero; when the density exceeded the environmental carrying capacity (i.e., 250 ind/ha), there was primarily emigration. Thus, the environmental carrying capacity was a crucial factor limiting the dispersal [40].

### 4.2. Pikas Managements

The campaign against plateau pika becomes more prevailing with the conduction of the ecological conservation and restoration project on the whole Tibetan Plateau since 2005. The principal method for reducing plateau pika abundance is lethal control using D-type Botulin [24]. However, the pikas' abundance could recover via once lethal control using poison [25]. It is urgent to determine an optimal approach to sustainably reduce pikas abundance.

To achieve better efficacy to control plateau pikas, lethal control programs are always conducted in spring or winter [3], the frequency of management programs primarily depends on the political task, not a scientific basis [25]. Here, we set different potential scenarios to analyze the population dynamics after lethal control. The lethal control conducted in April each year (Case 1) incontrovertibly reduces pikas' abundance over long term period. Though the statistically analysis suggested that the abundance under Case 2 was significantly lower than that of Case 0, the peak abundance exceeded 200 ind/ha in the year without poisoning. When lethal control was carried out in April with a 3-year interval (Case 3) or in October with a 2-year interval (Case 4), the abundances showed no significant differences with Case 0, population abundances recovered in the year without poisoning, and the peak abundances in June also exceeded 200 ind/ha. When the poisoning program was carried out in both April and October with a 2-year interval (Case 5), the population abundance was significantly lower than that of Case 0. In summary, only when poisoning programs were conducted once each year, or twice with 2-a year interval, can the pika population be controlled under reasonable abundance.

There are four approaches affecting population dynamics: Reproduction, death, immigration, and emigration [41]. In our model, reproduction parameters were stable, while mortality, immigration, and emigration were dependent by both abundance and environmental carrying capacity. The sustainable low abundances in the Case 1 and 5 suggested that frequent and continuous lethal control managements could reduce pika abundance under the threshold, when abundance and environmental carrying capacity did not work on the mortality and migration. However, in Case 2, 3, and 4, when pika abundances exceed the threshold, both density and environmental carrying capacity limited the survival and migration [42–44]. The abundance in spring is crucial for pika population recruitment, as adults are mainly the subjects for the subsequent reproduction [45]. When the lethal control area is not big enough, animals can disperse from outside [46]. Meantime, the mortality over summer is also vital for population recruitment, as the low density may allow the population's own high survival [27]. Under these circumstances, the population can recover [47–49].

*4.3. Management Implication*

This study demonstrates that abundance and environmental carrying capacity (i.e., habitat) impact mortality and dispersal, and finally influence the population dynamics after lethal control. To effectively reduce plateau pika abundance, restraining the mortality and dispersal is important, and there are several approaches that may be done: (i) Enhance the mortality of pests, via increasing the frequency of lethal control [50], or biological control, such as predators or parasites [4,51]; (ii) expand the regions of lethal control, to make it so pests could not disperse quickly [52]; (iii) build an isolation fence or physical barriers [53]; (iv) change the landscape to reduce the habitat suitability and environmental carrying capacity [54]; (v) reduce the reproduction via fertility control [55,56]. The best management decision for the control of small mammals may integrate different strategies and comprehensively manage their populations [57].

Our model performed well to simulate the population dynamics of plateau pikas in the study site, and found that a sustainable poisoning program should be conducted to sustainably control pika abundances. The model developed is based on the important assumptions which could be improved, that is primarily depending on the better knowledge of population demography (e.g., reproduction, mortality, migration, etc.). The model could be generalized to the management of other pests. Several factors should be considered: (1) The demographic parameters of the species from respective region, especially for those species with spatial-temporal variable parameters; (2) the specific effects of population density on recruitment; (3) the suitability of landscape or habitats and environmental carrying capacity; (4) the setting of management scenarios based on life history; (5) animal dispersal from the adjacent regions. Other factors, such as climate, human disturbance, or predation would also provide managers with insight into how controlling measures may impact pest populations prior to actual implementation.

**Supplementary Materials:** The following are available online at https://www.mdpi.com/2071-1050/13/2/543/s1. Table S1: Symbols used in model equations. Table S2: Description, values and sources of parameters used in the model.

**Author Contributions:** Conceptualization, J.Q.; methodology, J.Q., Z.L.; validation, J.Q., Z.L. and Y.L.; formal analysis, J.Q., Z.L.; investigation, J.Q.; writing—original draft preparation, J.Q.; writing—review and editing, Z.L., Z.G., Y.L., H.Z.; funding acquisition, J.Q. All authors have read and agreed to the published version of the manuscript.

**Funding:** This work was funded by the Second Tibetan Plateau Scientific Expedition and Research (STEP) program [2019QZKK0501], National Natural Science Foundation of China [31770459], Strategic Priority Research Program of the Chinese Academy of Sciences [XDA2002030302], CAS "Light of West China" Program, Youth Innovation Promotion Association CAS, Thousand People Plan of Qinghai Province, Joint Grant from Chinese Academy of Sciences-People's Government of Qinghai Province on Sanjiangyuan National Park [LHZX-2020-01], Qinghai Province Natural Sciences

Foundation [2021-ZJ-929], Qinghai Provincial Key R&D and Transformation Program [2019-SF-152], Construction Fund for Qinghai Key Laboratories (2021-ZJ-Y01).

**Institutional Review Board Statement:** Ethical review and approval were waived for this study, due to the data used were obtained from the references.

**Informed Consent Statement:** Not applicable.

**Data Availability Statement:** Data are contained within the article and supplementary materials.

**Acknowledgments:** We sincerely thank the editor and three anonymous reviewers for their valuable comments and suggestions.

**Conflicts of Interest:** The authors declare no conflict of interest.

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
