# Peer review of "A System Dynamics Model for Assessing the Efficacy of Lethal Control for Sustainable Management of Ochotona curzoniae on Tibetan Plateau"

_sustainability, doi:10.3390/su13020543_

Round 1
Reviewer 1 Report
Some specific comments:
Page 2, line 41: …”residual pests” instead of … “the residue pests” ..
Page 2, line 84: “Finally, we discussed” … instead of “Finally, we discuss” ..
Author Response
Dear reviewer:
Thank you for your comments concerning our manuscript entitled “Assessing the Efficacy of Lethal Control for Sustainable Pest Management on Tibetan Plateau”. We have studied the comments carefully and made corrections which we hope to meet with the approval. We make specific responses to the issues raised by each review, please see the details as followings.
Point 1: Page 2, line 41: …”residual pests” instead of … “the residue pests” . Response 1: thanks, we delete the “the” here in line 44.
Point 2: Page 2, line 84: “Finally, we discussed” … instead of “Finally, we discuss” .. Response 2: thanks, we delete this sentence according to two reviewers’ comments.
Thanks again for your comments.
Best wishes.

Reviewer 2 Report
This manuscript investigates the “Assessing the Efficacy of Lethal Control for Sustainable Pest Management on Tibetan Plateau”. The manuscript describes a system dynamics model is developed to parameterize a population model for this species. There is no clear conclusion from this work. Also, the manuscript needs re-analyzing some data besides revising statistical methods on some topics. Please see my specific comments below:
Ls.1-2: The title does not relate the study based on a system dynamics model to parameterize a population model. Change.
Ls.29-30: keywords serve to widen the opportunity to be retrieved from a database. To put words that already are into title and abstracts makes KW not useful. Please choose terms that are neither in the title nor in abstract.
L.35: …Pests transmit diseases…
L.36: Confuse, How pests destroy the stability of the ecosystem?
L.38: Delete “rapidly”
L.41: Again, delete “rapidly”
Ls.44-47: This paragraph does not have a logical sequence with the previous or next paragraph. Rewrite.
L.46: Delete “finally”
Ls.75-85: The main objective is unclear. Please, summarize and write an only objective.
L.96: Whats’s “K.” humilis? Also K. humilis should be in italic.
L.105: …20-30 adult female…Place one only number sample.
L.114: …11-25 adult females… Again, place one only number sample.
L.135: Delete “mainly”
Ls.148-150: Confuse, You write 3 cases but in the results, you show 5 or 6 possible cases. Revise and standardize for 5 or 6 cases throughout the manuscript.
L. 157: …(F1, 17 = 46.066, p < 0.001… his information can only be the result of an analysis of variance. You didn't mention this in the materials and methods.
Figure 2 is not relevant. Delete.
Figure 3: What exactly does figure 3 show?
Table 1: These data should be analyzed with poisson (for numbers) or binomial (for ratios) generalized linear models (GLM). In particular, the analyses of the number of cases and time should include line (year/months) and treatment (populations?), plus the interaction time x treatment as predictors.
Ls.268-278: Any conclusion? You have several results obtained in your investigation but you do not conclude anything. Please rewrite this paragraph.
Reviewer 3 Report
The article “Assessing the Efficacy of Lethal Control for Sustainable Pest Management on Tibetan Plateau” is very interesting, as the field work carried out allowed to compare the population trend of plateau pica recorded in the natural habitat with that obtained through a simulation model.
The work seems well conducted, but shows some concerns that need to be addressed. The paper can be considered for publication after minor revisions.
Main concerns:
Title and key words: the term “pest management” is quite general, as in title and key words the studied species is not mentioned, it should be specified that the paper regards a rodent pest, and the scientific name should be added.
Title could be “Assessing the Efficacy of Lethal Control for Sustainable Management of Ochotona curzoniae on Tibetan Plateau
Key words: “rodent pest management”
In M&M 2.2 authors wrote “The distance between two sites is about 12 km.” but the choice of two different study areas has not already mentioned, and it is clarified in the following paragraphs. Move the information about the distance between the sites (one used for natural population demography and one or two for control population demography) and clarify the correct number of study sites.
Further point by point comments are reported below.
Line 15: which threshold? Specify
Line 24: change “ selecting optimal strategies” with “in order to select optimal control strategies”
Line 33: into a new habitat
Line 34: causing adverse effects
Line 35: rodent pests transmit
Line 37: “acute toxicity and death”, death is not acute
Line 39: for its management
Line 41. The residual individuals or the surviving individuals
Line 46: change dynamics with abundance
Line 40: “Recently” the cited literature is 2007, better “in the last decades”
Line 53: There are 52 around 800 million small rodents belonging to about ….. species,
Line 55: “Plateau pika (Ochotona curzoniae Hodgson, Lagomorpha Ochotonidae)”; remove “which” or add “is”
Line 56: change occupying with representing
Line 58: “can reach more” remove “to”
Lines 59-60: “since it provides nests for small birds 59 and lizards, prey for predators” explain better
Line 63: “aggravating”
Line 68: remove “there were” before “around 400 million Yuan …”;
Line 69: “were used to poison”
Line 79: with the aim of planning optimal strategies
Line 89: “The distance between two sites is about 12 km.” the choice of two different study areas has not already mentioned, it is clarified in the following paragraph. Move the information about the distance between the sites (one used for natural population demography and one or two for control population demography) and clarify the correct number of study sites.
Line 96: K. humilis grassland: it seems that K. humilis is a plant species, write the complete name or clarify
Line 113: to assess the reproductive parameters
Line 123: how did you select an environmental carrying capacity of 250 ind/ha? If it derives from literature, cite it
Line 136 coincident instead of coincided
Line 137-138: unclear
Line 147: carried out
Results: the total number of caught/released pikas in the two study areas should be added, as well as the total number of dissected pikas
Paragraphs 3.1 and 3.2 have the same title. Par. 3.1 could be changed as follows: Plateau pika population trend under natural conditions
Line 156: showed a seasonal population trends consistent with …
Figure 3: it could be useful to add “Case 0”, Case 1” …. in each graph. Why did you report the horizontal line corresponding to 100 ind/ha? Does it represent a damage threshold? However, in the text the population density corresponding to the damage threshold is not reported. Explain and add more information about threshold density
Line 194: sustainably reduce pikas abundances: what do you mean for sustainably?
Line 203: it should be noted
Line 210: a dramatic population…
Lines 212-213: All paragraph about predation rate is unclear. The survival rates used in the model were obtained from the natural population and included the predation effect, therefore you did not consider in the model the effects of predation risk on the survival, is it correct? Please explain better.
Line 225: check for English language
Kind regards
Round 2
Reviewer 2 Report
The manuscript “A system dynamics model for assessing the Efficacy of Lethal Control for Sustainable Management of Ochotona curzoniae on Tibetan Plateau” has been improved and all my questions were taken into account. I recommend the publication in “Insects".